# Urinary 8-iso PGF$_{2\alpha}$ and 2,3-dinor-8-iso PGF$_{2\alpha}$ can be indexes of colitis-associated colorectal cancer in mice

Yusuke Miyazaki, Tatsuro Nakamura, Shinya Takenouchi, Akane Hayashi, Keisuke Omori, Takahisa Murata*

Department of Animal Radiology and Graduate School of Agricultural and Life Sciences, The University of Tokyo, Tokyo, Japan

* amurata@mail.ecc.u-tokyo.ac.jp

**Data Availability Statement:** All relevant data are within the manuscript and its Supporting Information files.

## Abstract

Early diagnosis of colorectal cancer is needed to reduce the mortal consequence by cancer. Lipid mediators play critical role in progression of colitis and colitis-associated colon cancer (CAC) and some of their metabolites are excreted in urine. Here, we attempted to find novel biomarkers in urinary lipid metabolite of a murine model of CAC. Mice were received single administration of azoxymethane (AOM) and repeated administration of dextran sulfate sodium (DSS). Lipid metabolites in their urine was measured by liquid chromatography mass spectrometry and their colon was collected to perform morphological study. AOM and DSS caused inflammation and tumor formation in mouse colon. Liquid chromatography mass spectrometry-based comprehensive analysis of lipid metabolites showed that cyclo-oxygenase-mediated arachidonic acid (AA) metabolites, prostaglandins, and reactive oxygen species (ROS)-mediated AA metabolites, isoprostanes, were predominantly increased in the urine of tumor-bearing mice. Among that, urinary prostaglandin (PG)E$_2$ metabolite tetranor-PGEM and PGD$_2$ metabolite tetranor-PGDM were significantly increased in both of urine collected at the acute phase of colitis and the carcinogenesis phase. On the other hand, two F$_2$ isoprostanes (F$_2$-IsoPs), 8-iso PGF$_{2\alpha}$ and 2,3-dinor-8-iso PGF$_{2\alpha}$, were significantly increased only in the carcinogenesis phase. Morphological study showed that infiltrated monocytes into tumor mass strongly expressed ROS generator NADPH (p22$^{phox}$). These observations suggest that urinary 8-iso PGF$_{2\alpha}$ and 2,3-dinor-8-iso PGF$_{2\alpha}$ can be indexes of CAC.

## Introduction

Six hundred million people die of colorectal cancer per year all over the world. Inflammatory bowel disease (IBD), which is characterized by relapse and remission of intestinal mucosal inflammation, is a major risk of colitis-associated colorectal cancer (CAC) [1]. Statistical research showed that 5-year survival rate of colorectal cancer is depending greatly on disease stage at diagnosis; 90% at localized stage and 14% at distant stage [2]. Thus, early diagnosis of colorectal cancer is critical to prolong lifespan for patients.

**Funding:** This work was supported by a Grant-in-Aid from the Japan Society for the Promotion of Science (19H03569, 17H01509, 18K08809 to TM, 18K14603 to TN); Shimadzu Science Foundation (to TM); Kobayashi Foundation (to TM); The Morinaga Foundation for Health & Nutrition (to TM); Takeda Science Foundation (to TM).

**Competing interests:** The authors have declared that no competing interests exist.

Sigmoidoscopy and colonoscopy are often performed for diagnosis of colorectal cancer [3]. These diagnostic methods are burdensome for patients, and they need well-experienced specialists and equipments to perform [4, 5]. Recent studies reported that serum micro RNA-141 or osteopontin [6, 7], microsatellite instability or sialyl-Tn antigen in biopsy specimens [8, 9] were useful for biomarkers of colorectal cancer. However, these procedures are invasive, and they also require special equipment. More convenient biomarkers for CAC are still required.

Researchers have used murine colorectal carcinogenesis models to investigate the mechanisms of the onset/progression of CAC. Azoxymethane (AOM)/dextran sulfate sodium (DSS)-induced colitis and CAC model has been often used because of its convenience and stability of incidence [10]. Previous studies showed that sustained and abundant infiltrations of neutrophils and macrophages were observed in the colonic mucosal tissue of AOM/DSS-treated mouse [11–13]. Other studies reported that reactive oxygen species (ROS) produced by infiltrating immune cells into lamina propria [14] damaged DNA, which in turn promotes carcinogenesis in AOM/DSS-treated mouse [15].

Lipid mediators are bioactive substances produced from polyunsaturated fatty acid of the cell membrane and they regulate inflammation and carcinogenesis. Cyclooxygenase (COX), lipoxygenase (LOX) and cytochrome P450 (CYP) mediate oxygenation of PUFA, such as arachidonic acid (AA), and synthesize lipid mediators. In human colon cancer, protein expressions of a COX isoform, COX-2, and a LOX isoform, 5-LOX, were increased [16]. Experimental studies showed that pharmacological inhibition of COX-2 ameliorated CAC induced by administration of AOM/DSS [17]. Gene deletion of microsomal prostaglandin (PG)E synthase inhibits the development of carcinogen-induced colon cancer [18]. On the other hand, the deficiency of PGD synthase aggravated AOM/DSS-induced colitis and CAC in mice [19]. These reports indicate that lipid mediators critically regulate the progression and onset of colitis/CAC.

Since lipid mediators and their related metabolites are excreted into urine, we generated AOM/DSS-induced colitis and CAC model mice for exploration of novel urinary index of CAC in this study and we discovered candidate substances for biomarker of CAC.

## Materials and methods

### Reagents

The following reagents were used: 6-keto $PGF_{1\alpha}$-$d_4$, thromboxane $(TX)B_2$-$d_4$, $PGF_{2\alpha}$-$d_4$, $PGE_2$-$d_4$, $PGD_2$-$d_4$, leukotriene $(LT)C_4$-$d_5$, $LTB_4$-$d_4$, 5(S) HETE-$d_8$, 12(S) HETE-$d_8$, 15(S) HETE-$d_8$, PAF C16-$d_4$, Oleoylethanolamide-$d_4$, tetranor-PGEM, tetranor-PGEM-$d_6$, tetaranor-PGDM, tetranor-PGDM-$d_6$, $LTE_4$, $LTE_4$-$d_5$, 11-dehydro-$TXB_2$, 11-dehydro-$TXB_2$-$d_4$, 8-iso $PGF_{2\alpha}$, 8-iso $PGF_{2\alpha}$-$d_4$, 2,3-dinor-8-iso $PGF_{2\alpha}$ (Cayman Chemical, USA); azoxymethane, DEXTRAN SULFATE SODIUM SALT REAGENT GRADE, ethanol, methanol, acetonitrile, hexane, acetic acid, formic acid, LabAssay™ Creatinine, hydrogen peroxide, citric acid monohydrate, trisodium citrate dehydrate, mayer's hematoxylin solution (FUJIFILM Wako, Japan); 3,3'-diaminobenzidine, tetrahydrochloride (DOJINDO, Japan); TritonX-100 (MP Biomedicals); normal donkey serum (Merck Chemicon, USA).

### Induction of colitis and CAC

Seven- to nine-weeks old female wild type C57BL/6 mice were used. All experimental procedures in this study were approved by the Institutional Animal Care and Use Committee at the University of Tokyo (P11-576 and P08-258). Mice were intraperitoneally administrated the large intestine-specific carcinogen AOM (12 mg/kg). Five days later, mice were provided 2% DSS in drinking water ad libitum for 4 days, followed by a 17-day-off period (Fig 1A). This

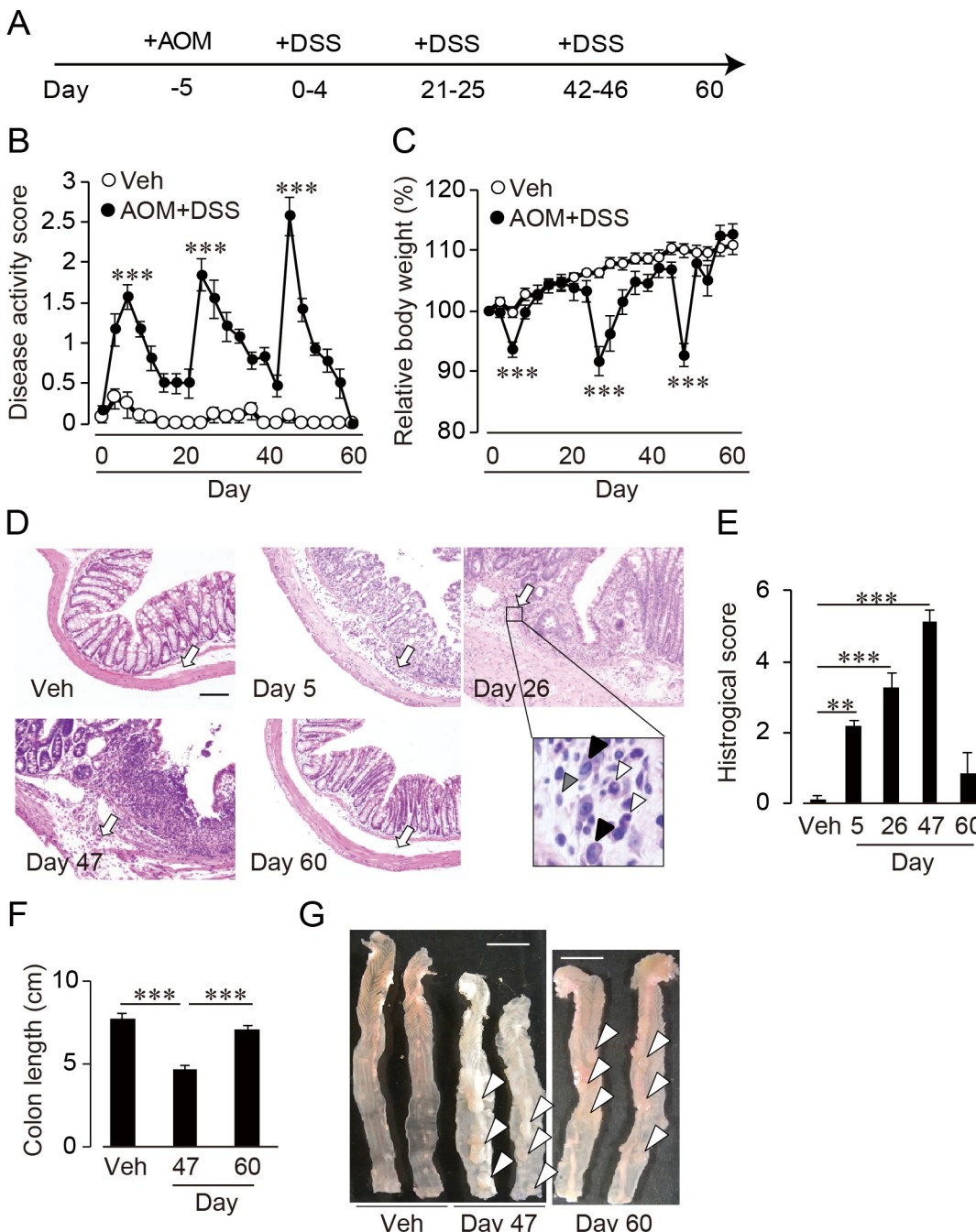

**Fig 1. AOM/DSS-induced colitis and CAC in mice.** (A) Schematic figure of AOM/DSS administration. (B) DAI and (C) body weight (n = 8–26). Body weight is showed as a value relative to the ratio of the body weight at day 0. (D) Representative histological images (hematoxylin & eosin staining) of colon on day 5, 26, 47 and 60 compared with vehicle treatment group. Black arrow, normal or damaged mucosa; white arrow, infiltrating cell; black arrowhead, macrophage; white arrowhead, neutrophil; grey arrowhead, lymphocyte. Scale bar, 100 εm. (E) Histological scoring of colon inflammation (n = 6–10). **, p<0.01; ***, p<0.001. (F) Colon length on day 47 and 60 compared with vehicle treatment group (n = 8–13) and (G) macroscopic images of a whole colon. White arrowhead, tumor. Scale bar, 1 cm. ***, p<0.001.

cycle was performed three times. Colon tissue was sampled at day 5, day 26, day 47 and day 60. No randomization was used to allocate mice to any groups and confounders were not controlled. The number of mice were 13 in vehicle group and 8–26 in AOM/DSS administration

group depending on the phase of day 5, 26, 47 and 60. Disease activity index (DAI) and body weight were checked every 3 days. The DAI was scored according to stool condition as followed: 0, normal; 1, soft but formed; 2, very soft; 3, diarrhea; 4, bloody diarrhea. These parameters were also used in previously published paper [19] and appropriate to assess animals' health and well-being.

### Histopathological assessment of colitis and CAC

Mice were properly euthanized by cervical dislocation. Colon tissue was fixed in 4% paraformaldehyde for 24 hours and embedded in paraffin. Tissue was sectioned at 4 μm and stained with hematoxylin and eosin in a basic protocol. For assessment of colitis severity, histological scoring was performed as shown in Table 1.

### Measurement of lipid metabolites

Urine was serially collected from the same mouse by metabolic cages and preserved at -80˚C. Sample solutions were prepared by mixing 100 μl urine, 850 μl deionized water, 50 μl internal standards (the composition is shown in S1 and S2 Tables) and 10 μl formic acid. Solutions were loaded onto methanol-conditioned and water-equilibrated solid-phase extraction (SPE) cartridges (Oasis HLB, Waters, USA). Followed by wash of cartridges by 5% (v/v) acetonitrile or water and hexane, the lipid metabolites absorbed to the cartridges were eluted with methanol and reconstituted in 80% (v/v) methanol. Samples were injected to LCMS-8030 (Shimadzu, Japan).

For comprehensive analysis, the liquid chromatographic separation was performed using a Phenomenex Kinetex C8 column (Shimadzu) and using a mobile phase consisting of 0.05% (v/v of water) formic acid (solvent A) and 0.05% (v/v of acetonitrile) formic acid (solvent B) (The gradient program is shown in S3 Table). The production amount of each lipid metabolites was determined in area under the curve (AUC).

For absolute measurement, the liquid chromatographic separation was performed using an Inertsil ODS-3 column (GL Sciences, Japan) and using a mobile phase consisting of 0.02% (v/v) acetic acid (solvent A) and acetonitrile (solvent B) (The gradient program is shown in S3 Table and other detailed settings are shown in S4 Table). Absolute value of 2,3-dinor-8-iso $PGF_{2\alpha}$ was adjusted for 8-iso $PGF_{2\alpha}$-$d_4$ as previously described [20]. The concentration value was calculated by correcting with the amount of creatinine in urine.

**Table 1. Histological scoring.**

| Cell infiltration | Score |
|---|---|
| Low frequency in the lamina propria | 0 |
| High frequency in the lamina propria | 1 |
| Extending into the submucosa | 2 |
| Transmural extension of the infiltration | 3 |
| Tissue damage | Score |
| No mucosal damage | 0 |
| Surface mucosal erosion | 1 |
| Focal ulceration | 2 |
| Extensive mucosal damage and extention into deeper structures of the bowel wall | 3 |

The extent of cell infiltration and tissue damage was assessed. The sum of these scores was used as an index of colon inflammation.

In some experiments, the concentrations of 8-iso $PGF_{2\alpha}$ and 2,3-dinor-8-iso $PGF_{2\alpha}$ were measured in plasma and colon tissues of naive or AOM+DSS-treated mice. Briefly, plasma was deproteinized by mixing with the same equivalent of organic solvent (metnaol: acetonitrile = 1:1, v/v) contaiting 5% 5N HCl. The colon tissues or polyp were homogenized in 300 μl methanol. After centrifusion, the supernatants were cleaned by SPE and the analytes were injected into LC-MS/MS, as descrived above. Because both 8-iso $PGF_{2\alpha}$ and 2,3-dinor-8-iso $PGF_{2\alpha}$ could not be detected in plasma by LCMS-8030, we utilized successor model LSMS-8060 to measure plasma and colonic concentrations of $F_2$-IsoPs. From the standard curve, the lower limit of quantification of 8-iso $PGF_{2\alpha}$ was 0.11 ng/ml and that of 2,3-dinor-8-iso $PGF_{2\alpha}$ was 0.19 ng/ml, respectively. The tissues concenctrations were expressed as pg per mg tissue weight.

### Immunostaining

Paraffin-embedded colon tissue sections were deparaffinized and treated with blocking reagent (5% normal donkey serum and 0.1% TritonX-100 in PBS) for 40 minutes at room temperature. The sections were incubated overnight at 4˚C with anti-COX-2 goat polyclonal antibody (diluted 1:100, Santa Cruz, USA), and then incubated with biotinylated anti-goat IgG horse antibody (diluted 1:200, VECTOR, USA) for 2 hours. Antibodies were visualized by treatment with hydrogen peroxide and 3,3'-diaminobenzidine, tetrahydrochloride. Counterstain was performed by hematoxylin. For p22[phox] staining, anti-p22[phox] mouse monoclonal antibody (diluted 1:200, Santa Cruz, USA) was applied as primary antibody. For ROS detection, freshly prepared frozen tissue sections were incubated with 2 μM dihydroethidium for 30 minutes at 37˚C.

### Statistical analysis

No criteria for including and excluding were set and no animals and data points were excluded. Results were expressed as the mean ± SEM. DAI and body weight were assessed by two-way ANOVA, followed by Bonferroni post-tests for comparison between more than two groups. Comprehensive analysis was assessed by the unpaired Student t test for comparison between two groups. Other data evaluations were conducted using one-way ANOVA, followed by Tukey test for comparison between more than two groups. We analyzed the result by GraphPad Prism version 3.02 (GraphPad Software, San Diego, California, USA, www.graphpad.com).

## Results

### Induction of colitis and CAC by administration of AOM/DSS

We generated colitis and CAC model by single injection of AOM followed by repeated cycles of DSS administration as shown in Fig 1A. Almost all of the animals didnot die unexpectedly. Vehicle group of mice drinking normal water did not represent any symptoms of colitis indicated as disease activity index (DAI, Fig 1B). Their body weight was gradually increased (Fig 1C). Administration of AOM/DSS increased DAI and decreased body weight of mice (Fig 1B and 1C). Their symptoms were exacerbated just after the beginning of each DSS administration and they gradually relieved after that (Fig 1B and 1C). On day 60; 14 days after the end of third administration of DSS, the disease scores subsided to the same level as vehicle group.

Histopathological study showed that vehicle group showed no mucosal damage of colon (representative pictures are shown in Fig 1D and histological scores are summarized in Fig 1E) and little infiltration of immune cells. Repeated cycles of DSS administration damaged colonic mucosa (shown by a black arrow) and infiltrated mononuclear cells and neutrophils into the

lamina propria. The severity of their manifestations was increased in DSS administration cycle-dependent manner (Fig 1D and 1E). The colon length was strongly shortened in the phase of third DSS administration (Fig 1F, day 47). In contrast, 14 days after the last DSS administration (day 60), mucosal damage and infiltration of immune cells were almost completely disappeared and the histological score and the colon length were recovered to the level of vehicle treatment (Fig 1D–1F, day 60).

On the third DSS administration cycle, some tumors were formed in the distal area of colon (Day 47, Fig 1D, black arrow and Fig 1F). We also confirmed the presence of tumors in colon by morphological study (Fig 1G). The tumors resided in the colon even after the remission of inflammation (day 60). Thus, the AOM/DSS administrations induced CAC after day 47.

## Comprehensive analysis of lipid metabolites in urine

To explore the candidates of urinary indexes of CAC, we performed comprehensive analysis of lipid metabolites excreted in the urine of AOM/DSS-treated mice on day 60. We detected 32 lipid metabolites in 131 types of that we measured (Fig 2 and S5 Table, which shows detected lipid mediators). Among them, arachidonic acid (AA)-derived metabolites accounted for 88% (28 types) of the detected metabolites (Fig 2 and S5 Table). The amounts of 10 lipid metabolites were significantly increased in the urine of AOM/DSS administrated mice compared with that of vehicle treated mice (Fig 2).

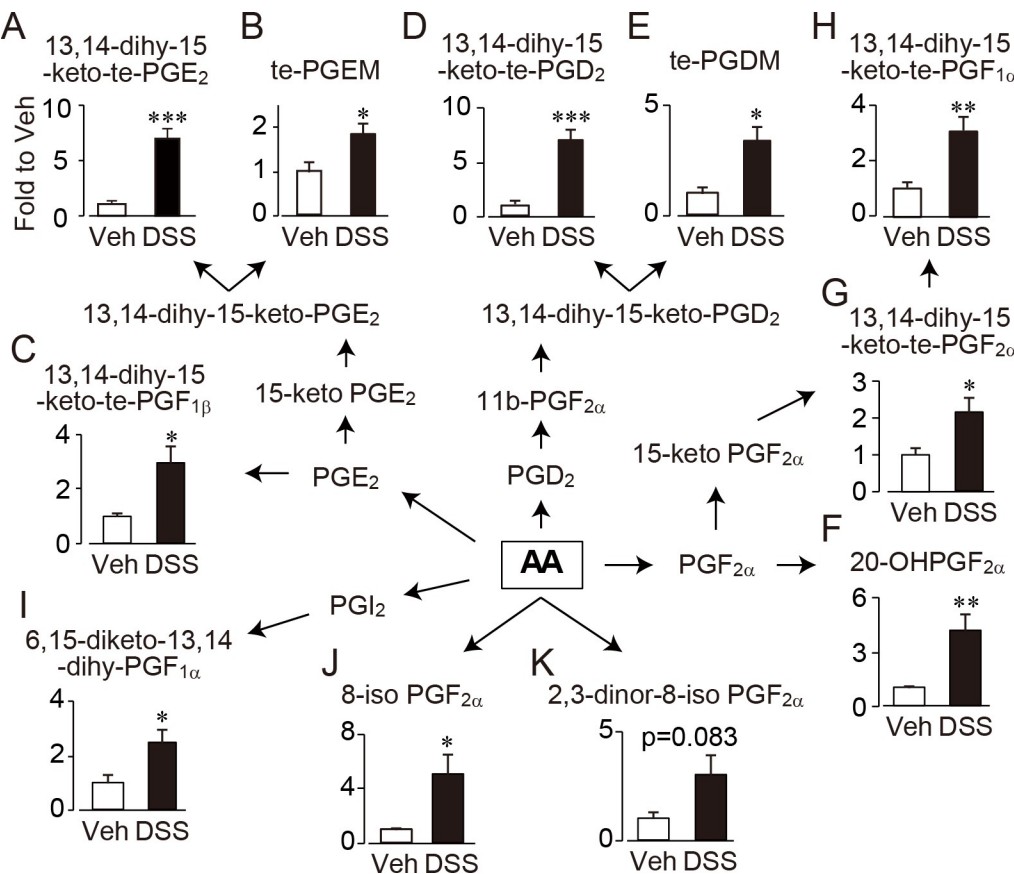

**Fig 2. Increased urinary lipid metabolites in CAC mice.** Lipid metabolites were comprehensively measured in the urine of vehicle or DSS administrated mice (n = 7 each). dihy, dihydro; te, tetranor; PG, prostaglandin; TX, thromboxane; AA, arachidonic acid. *, $p < 0.05$; **, $p < 0.01$; ***, $p < 0.001$.

Upon inflammation, PGs are synthesized from AA by COX, mainly COX-2, activity. Three types of $PGE_2$ metabolites (Fig 2A–2C), two types of $PGD_2$ metabolites (Fig 2D and 2E) and three types of $PGF_{2\alpha}$metabolites (Fig 2F–2H) were significantly increased in the urine of CAC-bearing mice. A metabolite of $PGI_2$ metabolite, 6,15-diketo-13,14-dihydro-$PGF_{1\alpha}$ (Fig 2I) was also significantly increased in the CAC mice urine. The other enzymatic oxidative products, AA-LOX metabolite; 18-carboxy-dinor-$LTB_4$ (S5 Table, p = 0.009) and EPA-CYP metabolite; 17,18-DiHETE (S5 Table, p = 0.04) were significantly increased in the urine of CAC-bearing mice. In addition to oxidative enzyme, ROS catalyzes non-enzymatic oxidation of AA and produces isoprostanes [21]. 8-iso $PGF_{2\alpha}$ (Fig 2J) was significantly increased in the CAC mice urine. A metabolite of 8-iso $PGF_{2\alpha}$, 2,3-dinor-8-iso $PGF_{2\alpha}$ tended to be increased in the CAC-bearing mice urine (p = 0.083, Fig 2K). From these results, we focused on AA-derived COX-2 and/or ROS metabolites as candidates of urinary CAC index.

## Expression of COX-2 and p22$^{phox}$ in mucosal tissue and tumor tissue

We next assessed the expression of COX-2 and a main subunit of NADPH oxidase (NOX), p22$^{phox}$ in the AOM/DSS stimulated colon tissues. As previously reported [17, 22], COX-2 was expressed in the inflamed colonic epithelial cells on day 5, 26 and 47 (Fig 3A). It was highly expressed in the inflamed colonic epithelial cells on day 5 and colorectal tumor cells in carcinogenesis period (Fig 3A).

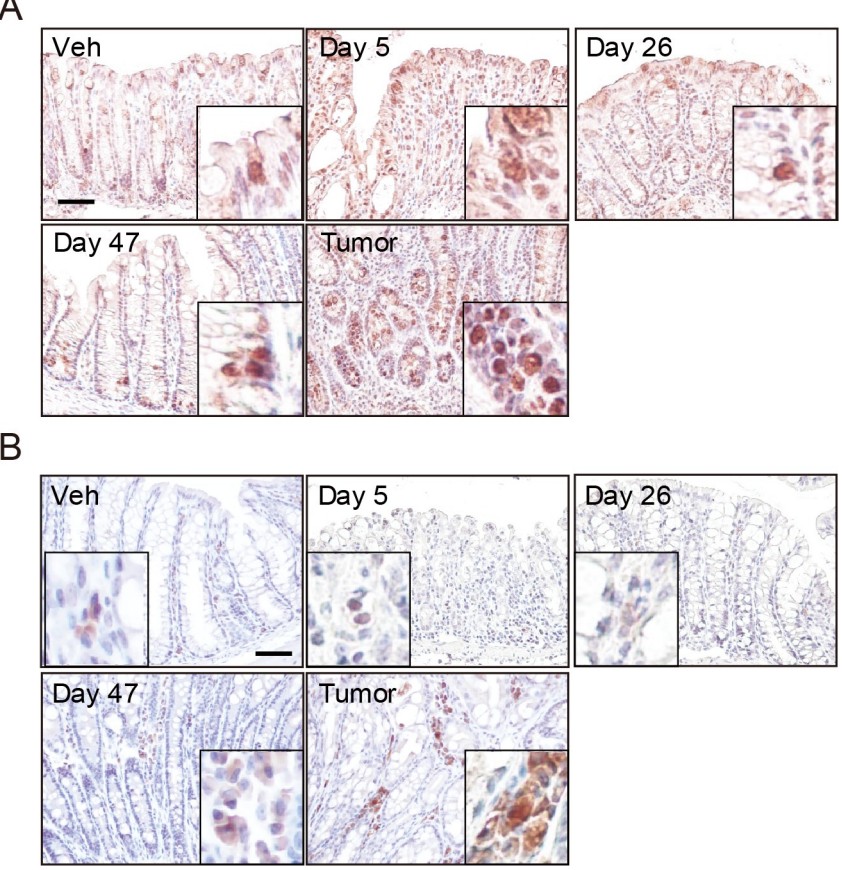

**Fig 3. The colon expression of enzyme related lipid metabolite production.** Representative images of immunohistochemistry of (A) COX-2 and (B) p22$^{phox}$. Areas surrounded by squares indicate COX-2 or p22$^{phox}$-positive cells. Scale bar, 50 μm.

A NOX isoform NOX1 was reported to be expressed in human colonic epithelial cells and colorectal cancer cells [23] and another NOX isoform NOX2 was expressed in innate immune cells, such as neutrophils and macrophages [24]. As shown in Fig 3B, the protein expression of p22$^{phox}$ was observed in the mononuclear cells infiltrated into lamina propria during DSS administration cycle and which was higher in the colorectal tumor tissue (day 60) than inflamed mucosal tissue (day 5–47, Fig 3B).

## Absolute measurement of lipid metabolites in the development of CAC

Both tetranor-PGEM and tetranor-PGDM are abundant urinary metabolites of $PGE_2$ or $PGD_2$, respectively [25, 26]. In addition, it is reported that two types of $F_2$-isoPs, 8-iso $PGF_{2\alpha}$ and 2, 3-dinor-8-iso $PGF_{2\alpha}$, are detectable in several biological fluids [27]. We next measured the absolute concentration of tetranor-metabolites and $F_2$-isoPs. Both tetranor-metabolites transiently increased on the acute phase of colitis (day 1–2) and carcinogenesis period (day 47, 60) (Fig 4A and 4B). On the other hand, the concentrations of $F_2$-isoPs were significantly increased only in the carcinogenesis period (Fig 4C and 4D).

To verify that 8-iso $PGF_{2\alpha}$ and 2, 3-dinor-8-iso $PGF_{2\alpha}$ were increased in the carcinogenesis period, we measured both $F_2$-IsoPs concentrations in plasma and colonic tissues. The levels of 8-iso $PGF_{2\alpha}$ in all plasma samples and that of 2, 3-dinor-8-iso $PGF_{2\alpha}$ in some samples were below the lower limit of quantification (LLOQ). Mean concentrations of 2, 3-dinor-8-iso $PGF_{2\alpha}$ in plasma samples (n = 3: control, n = 6: CAC) were comparable between control and CAC mice (Fig 4E). The levels of 8-iso $PGF_{2\alpha}$ in some colon tissues and polyp were also below the LLOQ (n = 1/4: control, n = 3/8: inflamed colon, n = 4/7: polyp). Unexpectedly, mean concentrations of 8-iso $PGF_{2\alpha}$, except below the LLOQ, were comparable between control colon, inflamed colon tissues and polyp (Fig 4F). The levels of 2, 3-dinor-8-iso $PGF_{2\alpha}$ in all colon samples were below the LLOQ.

## Discussion

In this study, we performed qualitative and quantitative analysis of the urinary lipid metabolites in AOM/DSS-induced colitis and CAC model mice. We found that the urinary levels of 8-iso $PGF_{2\alpha}$ and 2,3-dinor-8-iso $PGF_{2\alpha}$ were significantly increased in the phase of CAC. Considering that 2,3-dinor-8-iso $PGF_{2\alpha}$ is a metabolite of 8-iso $PGF_{2\alpha}$and is chemically stable, 2,3-dinor-8-iso $PGF_{2\alpha}$ would be an optimal candidate of urinary biomarker of CAC.

The significant increases of $F_2$-isoPs in urine were observed in the phase of tumor development but not acute/chronic inflammation. NADPH oxidase (NOX) is a group of membrane-associated enzyme, consisting of two membrane protein p22 and gp91 and catalyze one-electron reduction of oxygen [28]. As a result of this reaction, ROS is generated, and oxidative stress occurs due to disturbing redox balance by overproduced ROS. Overproduced ROS caused DNA damage and genetic mutation which leaded carcinogenesis *in vivo* [15]. At the same time, ROS mediates metabolism of AA to isoprostanes including 8-iso $PGF_{2\alpha}$ via free radical-catalyzed peroxidation [21]. Previous study has demonstrated that myeloid derived suppressor cells (MDSCs) from tumor-bearing mice expressed the enhanced levels of NOX subunits, such as gp91 and p22, and produced more ROS than that from tumor-free mice [29]. In addition, as previous report showing that MDSCs massively accumulated in the CAC region [30], we also observed a NOX subunit p22$^{phox}$-positive cells in the colonic lamina propria of mice with polyp. Thus, phase-dependent increase of $F_2$-isoP extraction would be considerd by NOX-expressed MDSC accumulation in CAC region. However, we colud not find the increased levels of $F_2$-IsoP in plasma and CAC region in this study. This may be because the concentration of $F_2$-IsoP produced in local region by ROS-generating cells was too small to

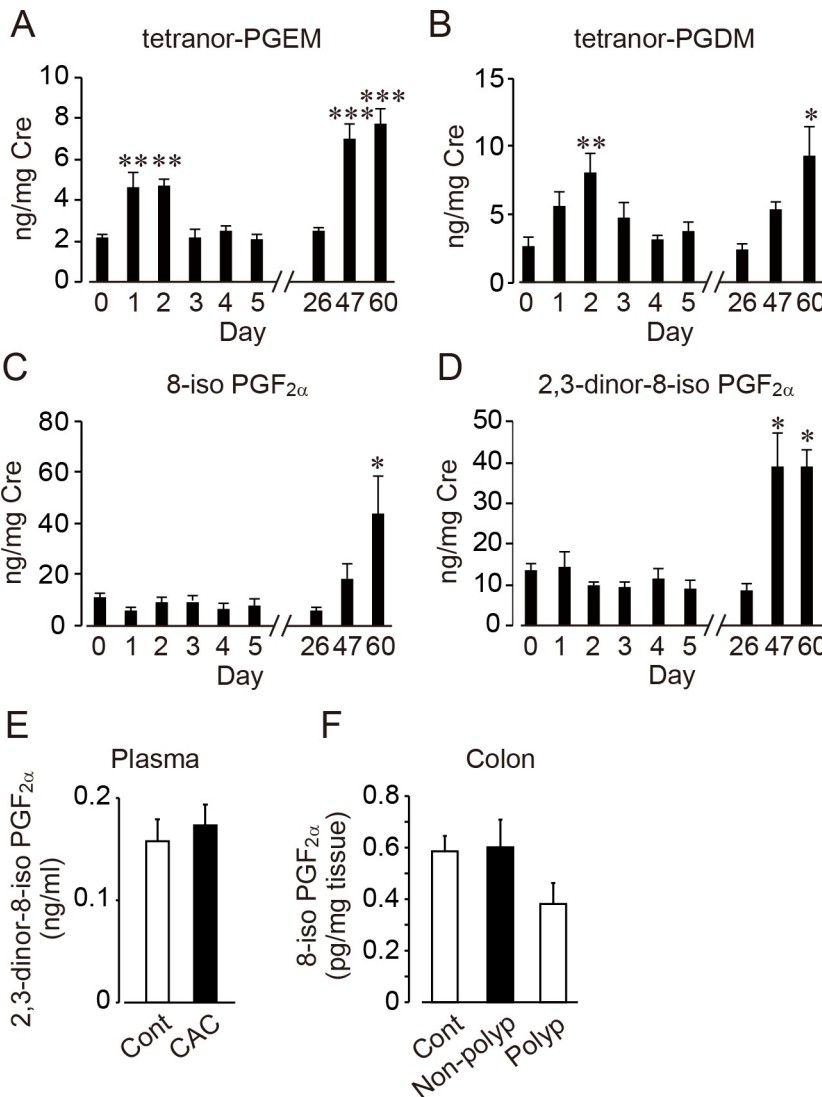

**Fig 4. The absolute concentration of lipid metabolites.** The urinary concentration of tetranor-PGEM (A), tetranor-PGDM (B), 8-iso $PGF_{2\alpha}$ (C) and 2,3-dinor-8-iso $PGF_{2\alpha}$ (D) in acute colitis and tumor formation period (n = 6–17). The concentration of creatinine in urine was measured to calculate the correction value. (E) Plasma levels of 8-iso $PGF_{2\alpha}$ (n = 3: control, n = 6: CAC). (F) The levels of 2,3-dinor-8-iso $PGF_{2\alpha}$ in colon tissues and polyp (n = 3: control, n = 5: inflaimed colon tissue without polyp, n = 3: polyp). The lower limit of quantification (LLOQ) of 8-iso $PGF_{2\alpha}$ or 2,3-dinor-8-iso $PGF_{2\alpha}$ were 0.19 or 0.11 ng/ml, respectively. The values below LLOQ were excluded from obtained data (n = 4: control, n = 8: CAC). *, $p < 0.05$; **, $p < 0.01$; ***, $p < 0.001$ compared with day 0.

quantify in mice. Indeed, the levels of 8-iso $PGF_{2\alpha}$ were below the LLOQ in all plasma samples and some colon samples. More sensitive methods are needed for measuring local $F_2$-IsoPs levels.

The concentration of tetranor-PGEM and tetranor-PGDM were significantly increased in the urine of early phase of colitis inflammation and CAC. It is reported that $PGE_2$ synthase m-PGES was expressed higher in tumor stroma cells of AOM-induced CAC model mice [18]. Urinary excretion of tatranor-PGEM was increased in patients suffering from ulcerative colitis [31] and colorectal cancer [32]. The expressions of two types of PGD synthase H-PGDS and L-PGDS are increased in the inflamed intestinal tumor tissue in mice and human [19, 33]. It is

reasonable that urinary concentration of major $PGE_2$ or $D_2$ metabolites is high in mice with colitis and CAC in this study. On the other hand, there are numerous studies showing the production and contribution of $PGE_2$ or $PGD_2$ in various types of inflammatory diseases other than CAC, such as viral-induced fever, food allergy, cystic fibrosis and lung metastasis [34–37]. Thus, urinary tetranor-PGEM and/or tetranor-PGDM may lack a specificity as a biomarker of CAC although it can be utilized as a supportive index reflecting inflammatory condition.

In the present study, we found candidates of urinary biomarker of CAC using AOM/DSS-induced CAC mouse model which represents similar characteristic with human CAC; diseased area of colon, infiltration of granulocytes, accumulation of β-catenin and mutation of K-Ras gene [38, 39]. However, it is still required to investigate the clinical usefulness of the urinary CAC marker found in this study. Further investigations using human urine sample are necessary to confirm this issue.

In conclusion, we identified urinary 8-iso $PGF_{2\alpha}$ and 2,3-dinor-8-iso $PGF_{2\alpha}$ as candidates of CAC biomarkers using murine colitis and CAC model. In addition, we found that urine is more appropriate for detecting CAC biomarker than plasma. Our findings may contribute to the improvement of early diagnosis of CAC.

## Supporting information

**S1 Table. The composition of internal standards mixture for comprehensive analysis.** (DOCX)

**S2 Table. The composition of internal standards mixture for absolute measurements.** (DOCX)

**S3 Table. Gradient program for comprehensive analysis and absolute measurement.** (DOCX)

**S4 Table. Ion mode, m/z value and elution time of each substance for absolute measurement.** Ion mode was selected fromnegative (-) or positive (+). (DOCX)

**S5 Table. Lipid metabolites detected in the urine of CAC mice model.** The values are shown as fold increase ± SE compared with the average of vehicle treated mice. DGLA, dihomo-γ-linoleic acid; AA, arachidonic acid; EPA, eicosapentaenoic acid; COX, cyclooxygenase; LOX, lipoxygenase; CYP, cytochrome P450; PG, prostaglandin; TX, thromboxane; LT, leukotriene; DiHETE, dihydroxyeicosatetraenoic acid; OxoEDE, oxoeicosadienoic acid; -, non-enzymatic oxidation. (DOCX)

## Author Contributions

**Conceptualization:** Tatsuro Nakamura, Takahisa Murata.

**Data curation:** Yusuke Miyazaki, Tatsuro Nakamura, Shinya Takenouchi, Akane Hayashi, Keisuke Omori.

**Formal analysis:** Yusuke Miyazaki, Tatsuro Nakamura, Keisuke Omori.

**Funding acquisition:** Takahisa Murata.

**Investigation:** Yusuke Miyazaki, Tatsuro Nakamura, Shinya Takenouchi, Akane Hayashi.

**Methodology:** Tatsuro Nakamura, Keisuke Omori, Takahisa Murata.

**Project administration:** Takahisa Murata.

**Resources:** Takahisa Murata.

**Software:** Takahisa Murata.

**Supervision:** Takahisa Murata.

**Validation:** Tatsuro Nakamura, Takahisa Murata.

**Writing – original draft:** Yusuke Miyazaki, Takahisa Murata.

**Writing – review & editing:** Tatsuro Nakamura, Shinya Takenouchi, Akane Hayashi, Keisuke Omori, Takahisa Murata.

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
