## [Decision Letter · Decision Letter 0]

3 Sep 2020

PONE-D-20-25696

Urinary 8-iso-PGF2α and 2,3-dinor-8-iso-PGF2α can be indexes of colitis-associated colorectal cancer in mice Lipid mediators as biomarker of colitis-associated colorectal cancer

PLOS ONE

Dear Dr. Murata

Thank you for submitting your manuscript to PLOS ONE. After careful consideration, we feel that it has merit but does not fully meet PLOS ONE’s publication criteria as it currently stands. Therefore, we invite you to submit a revised version of the manuscript that addresses the points raised during the review process.

While the reviewer appreciates the importance of the study, the reviewer has several concerns that need to be addressed before publication. Specifically, the cellular source of PGF2alpha and its metabolites is not directly investigated in the study. Hence, the authors need to investigate whether colon cancer cells or cancer tissues produce PGF2alpha and its metabolites. Moreover, it would be intrigued to determine the concentrations of PGF2alpha and its metabolites in the plasma of untreated and AOM/DSS-treated mice.

We look forward to receiving your revised manuscript.

Kind regards,

Hiroyasu Nakano, M.D., Ph.D.

Academic Editor

PLOS ONE

Journal Requirements:

2. As part of your revisions, please provide additional details pertaining to your animal research procedures. Please revise your Methods to address the following points: (1) sample size: how many animals per group and how did you determine the numbers (power analysis? pilot study? previously published data/findings?); (2) provide complete information about all methods undertaken to minimize pain and distress of the animals in your work. (3) Please discuss your monitoring parameters (physical and behavioral signs to assess health and well-being), in addition to your humane endpoints (criteria used to determine when to euthanize animals in cases where animals become sick/moribund). (4) Please state the rate of mortality for animals who died unexpectedly (and state cause(s) of death). (6) Please also discuss supportive care that you provided to the animals. (7) Lastly, please complete and submit the ARRIVE Guidelines checklist (Essential 10) with your revision: https://arriveguidelines.org/resources/author-checklists

Reviewers' comments:

Reviewer's Responses to Questions

**Comments to the Author**

1. Is the manuscript technically sound, and do the data support the conclusions?

Reviewer #1: Partly

2. Has the statistical analysis been performed appropriately and rigorously? 

Reviewer #1: No

3. Have the authors made all data underlying the findings in their manuscript fully available?

Reviewer #1: Yes

4. Is the manuscript presented in an intelligible fashion and written in standard English?

Reviewer #1: Yes

5. Review Comments to the Author

Reviewer #1: Miyazaki et al established a murine model of inflammation-induced colon cancer (CAC), and searched for novel lipid biomarkers in the urine of this model mouse. They identified contents of several metabolites derived from arachidonic acid are increased in the CAC urine. Some of them (PGD and E metabolites) are also increased in the inflammation period, and not candidates for biomarkers. They finally found that 8-iso-PGF2αand 2,3-dinor-8-iso-PGF2α are increased only in the CAC period, and concluded that these two PGF2 metabolites are candidate biomarkers for colorectal cancer. They also stained the mouse cancer tissues and showed that the expression of COX-2 and NADPH oxidase is increased in the cancer cells, and this can explain the increase of the metabolites.

This is an interesting study that utilizes a mouse oncogenic model to find biomarkers. The biosynthesis and metabolic inactivation of arachidonic acid-derived lipid mediators are conserved among various animal species, this strategy is reasonable. This group has a long history of measuring PG and PG metabolites, and the manuscript is simply and clearly written. The reviewer, however, has to raise several important issues to be addressed in the revised manuscript.

Major points

The authors claim that 8-iso-PGF2alpha and 2,3-dinor-8-iso-PGF2 are possible biomarkers based on the specific increase of these metabolites in CAC phase. They did not succeed in convincing the reviewer that the colon cancer cells can generate and metabolize PGF2 alpha. Quantification of PGF2alpha and its metabolites in the cancer tissues, hopefully, lazar-dissected tissues, is required. If they think that cancer cells produce PGF2alpha, which enters into the blood and is metabolized in the other tissues, they should measure PGF2alpha and metabolites in the plasma.

Regarding the application to the human situation, they should measure these metabolites in the urine of human colon cancer patients and show some representative results.

Minor points

Statistic analyses are missing in Fig. 1E, F, and Fig. 2.

6. PLOS authors have the option to publish the peer review history of their article (what does this mean?). If published, this will include your full peer review and any attached files.

Reviewer #1: No

---

## [Author Response · Author response to Decision Letter 0]

21 Dec 2020

Responses to the reviewers’ comments of the manuscript “Urinary 8-iso-PGF2α and 2,3-dinor-8-iso-PGF2 can be indexes of colitis-associated colorectal cancer in mice”.

Comment 1-1. Please ensure that your manuscript meets PLOS ONE's style requirements, including those for file naming. The PLOS ONE style templates can be found at

<Response>

We revised the manuscript and supplemental files according to the PLOS ONE style templates and they correctly meet PLOS ONE’s requirement. 

Comment 1-2 As part of your revisions, please provide additional details pertaining to your animal research procedures. Please revise your Methods to address the following points: (1) sample size: how many animals per group and how did you determine the numbers (power analysis? pilot study? previously published data/findings?); 

<Response>

We added the descriptions in manuscript in line 89-91 for (1); “The number of mice were 13 in vehicle group and 8 – 26 in AOM/DSS administration group depending on the phase of day 5, 26, 47 and 60.”

Comment 1-3 (2) provide complete information about all methods undertaken to minimize pain and distress of the animals in your work. (3) Please discuss your monitoring parameters (physical and behavioral signs to assess health and well-being), in addition to your humane endpoints (criteria used to determine when to euthanize animals in cases where animals become sick/moribund). (4) Please state the rate of mortality for animals who died unexpectedly (and state cause(s) of death). (6) Please also discuss supportive care that you provided to the animals. 

<Response>

We used eight weeks old C57BL/6J male mice (20-22 g). Mice were housed under standard laboratory conditions (22±1°C, 12-hour light/dark cycles, food and water ad libitum). Animal care and handling procedures were in accordance with International Association for the Study of Pain (IASP) guidelines for the use of animals in pain research and the protocols for animal care and use were approved by the Institutional Animal Care and Use Committee at the University of Tokyo (P11-576 and P08-258). All effort was made to limit the number of animals used. We set humane endpoint to prevent or alleviate pain and/or distress when 20% body weight loss, disability in food intake and water drinking are observed. However, all the mice used in the current study did not exceed the criteria and did not die unexpectedly. We revised the manuscript as follows:

Line 108 for (2); “Mice were properly euthanized by cervical dislocation.”

Line 93 for (3); “These parameters were also used in previously published paper [19] and appropriate to assess animals’ health and well-being.”

Line 173 for (4); “Almost all of the animals have not died unexpectedly.”

Line xx for (6); 

Comment 1-4 (7) Lastly, please complete and submit the ARRIVE Guidelines checklist (Essential 10) with your revision: https://arriveguidelines.org/resources/author-checklists

<Response>

We filled in the ARRIVE Guidelines checklist. And we added the following descriptions in manuscript; “Seven- to nine-weeks old female wild type C57BL/6 mice were used.” In line 82; “No randomization was used to allocate mice to any groups and confounders were not controlled.” In line 88; “No criteria for including and excluding were set and no animals and data points were excluded.” in line 161.

Reviewer #1: 

Comment 2-1 Miyazaki et al established a murine model of inflammation-induced colon cancer (CAC), and searched for novel lipid biomarkers in the urine of this model mouse. They identified contents of several metabolites derived from arachidonic acid are increased in the CAC urine. Some of them (PGD and E metabolites) are also increased in the inflammation period, and not candidates for biomarkers. They finally found that 8-iso-PGF2αand 2,3-dinor-8-iso-PGF2α are increased only in the CAC period, and concluded that these two PGF2 metabolites are candidate biomarkers for colorectal cancer. They also stained the mouse cancer tissues and showed that the expression of COX-2 and NADPH oxidase is increased in the cancer cells, and this can explain the increase of the metabolites.

This is an interesting study that utilizes a mouse oncogenic model to find biomarkers. The biosynthesis and metabolic inactivation of arachidonic acid-derived lipid mediators are conserved among various animal species, this strategy is reasonable. This group has a long history of measuring PG and PG metabolites, and the manuscript is simply and clearly written. The reviewer, however, has to raise several important issues to be addressed in the revised manuscript.

Major points

The authors claim that 8-iso-PGF2alpha and 2,3-dinor-8-iso-PGF2 are possible biomarkers based on the specific increase of these metabolites in CAC phase. They did not succeed in convincing the reviewer that the colon cancer cells can generate and metabolize PGF2 alpha. Quantification of PGF2alpha and its metabolites in the cancer tissues, hopefully, lazar-dissected tissues, is required. If they think that cancer cells produce PGF2alpha, which enters into the blood and is metabolized in the other tissues, they should measure PGF2alpha and metabolites in the plasma.

<Response>

Thank you. 8-iso-PGF2� is an oxidized metabolite of arachidonic acid not PGF2��and 2,3-dinor-8-iso-PGF2� is a metabolite of 8-iso-PGF2�. As following your suggestion, we measured the concentrations of these two metabolites in plasma and polyps of AOM+DSS-treated mice (day 60). As shown in below Figure, the levels of 2, 3-dinor-8-iso PGF2α (left figure) in plasma and that of 8-iso-PGF2α (right figure) in colon tissues were unexpectedly comparable between control and CAC mice. The concentrations of 8-iso-PGF2α in all plasma samples and that of 2, 3-dinor-8-iso PGF2α in all colon samples were below the lower limit of quantification (LLOQ). From these results, we could not find the increase levels of F2-IsoPs in plasma and CAC region in this study. This may be because the local production of 8-iso-PGF2� by ROS-generating cells was too small to quantify in mice. Indeed, the levels of 8-iso-PGF2α in some colon tissues and polyp were below the LLOQ (n=1/4: control, n=3/8: inflamed colon, n=4/7: polyp). It needs more sensitive methods for measuring local F2-IsoPs levels. In other words, urine is more appropriate for detecting CAC biomarkers than plasma. We added these results as Figure 4E-F and added the descriptions in the material and method (line 137-146), results (line 255-264), and discussion section (line 298-303).

Comment 2-2 Regarding the application to the human situation, they should measure these metabolites in the urine of human colon cancer patients and show some representative results.

<Response>

We attempted to obtain urine sample from patients in cooperating with Prof. Mamoru Watanabe, predecessor of Advanced Clinical Center for IBD in Tokyo Medical and Dental University Hospital in Japan. However, we could not obtain the data for this manuscript because it will take time to approve clinical studies and sampling. Please understand the situation.

Comment 2-3 

Statistic analyses are missing in Fig. 1E, F, and Fig. 2.

<Response>

We performed statistical analysis and added in Fig. 1E, F, and Fig.2.

---

## [Decision Letter · Decision Letter 1]

26 Dec 2020

Urinary 8-iso PGF2α and 2,3-dinor-8-iso PGF2α can be indexes of colitis-associated colorectal cancer in mice

PONE-D-20-25696R1

Dear Dr. Murata

We’re pleased to inform you that your manuscript has been judged scientifically suitable for publication and will be formally accepted for publication once it meets all outstanding technical requirements.

Kind regards,

Hiroyasu Nakano, M.D., Ph.D.

Academic Editor

PLOS ONE

Additional Editor Comments (optional):

Reviewers' comments:

Reviewer's Responses to Questions

**Comments to the Author**

1. If the authors have adequately addressed your comments raised in a previous round of review and you feel that this manuscript is now acceptable for publication, you may indicate that here to bypass the “Comments to the Author” section, enter your conflict of interest statement in the “Confidential to Editor” section, and submit your "Accept" recommendation.

Reviewer #1: All comments have been addressed

2. Is the manuscript technically sound, and do the data support the conclusions?

Reviewer #1: Yes

3. Has the statistical analysis been performed appropriately and rigorously? 

Reviewer #1: Yes

4. Have the authors made all data underlying the findings in their manuscript fully available?

Reviewer #1: Yes

5. Is the manuscript presented in an intelligible fashion and written in standard English?

Reviewer #1: Yes

6. Review Comments to the Author

Reviewer #1: (No Response)

7. PLOS authors have the option to publish the peer review history of their article (what does this mean?). If published, this will include your full peer review and any attached files.

Reviewer #1: No

---

## [Editor Report · Acceptance letter]

18 Jan 2021

PONE-D-20-25696R1 

Urinary 8-iso PGF_2α_ and 2,3-dinor-8-iso PGF_2α_ can be indexes of colitis-associated colorectal cancer in mice 

Dear Dr. Murata:

I'm pleased to inform you that your manuscript has been deemed suitable for publication in PLOS ONE. Congratulations! Your manuscript is now with our production department. 

Kind regards, 

on behalf of

Professor Hiroyasu Nakano 

Academic Editor

PLOS ONE